# Combining CSF and Serum Biomarkers to Differentiate Mechanisms of Disability Worsening in Multiple Sclerosis

**DOI:** 10.3390/ijms26146898

**Published:** 2025-07-18

**Authors:** Enric Monreal, José Ignacio Fernández-Velasco, Susana Sainz de la Maza, Mercedes Espiño, Noelia Villarrubia, Ernesto Roldán-Santiago, Yolanda Aladro, Juan Pablo Cuello, Lucía Ayuso-Peralta, Alexander Rodero-Romero, Juan Luís Chico-García, Fernando Rodríguez-Jorge, Ana Quiroga-Varela, Eulalia Rodríguez-Martín, Belén Pilo de la Fuente, Guillermo Martín-Ávila, María Luisa Martínez-Ginés, José Manuel García-Domínguez, Lluïsa Rubio, Sara Llufriu, Manuel Comabella, Xavier Montalban, Gary Álvarez-Bravo, José Luís Veiga-González, Jaime Masjuan, Lucienne Costa-Frossard, Luisa María Villar

**Affiliations:** 1Department of Neurology, Hospital Universitario Ramón y Cajal, Red Española de Esclerosis Múltiple (REEM), Red de Enfermedades Inflamatorias (REI), IRYCIS, Universidad de Alcalá, 28034 Madrid, Spain; enricmonreal@outlook.com (E.M.);; 2Department of Immunology, Hospital Universitario Ramón y Cajal, Red Española de Esclerosis Múltiple (REEM), Red de Enfermedades Inflamatorias (REI), IRYCIS, Universidad de Alcalá, 28034 Madrid, Spain; 3Department of Neurology, Hospital Universitario Getafe, Universidad Europea de Madrid, 28905 Madrid, Spain; 4Department of Neurology, Hospital Universitario Gregorio Marañón, 28007 Madrid, Spain; 5Department of Neurology, Hospital Universitario Príncipe de Asturias, 28805 Alcalá de Henares, Spain; 6Neuroimmunology and Multiple Sclerosis Unit, Neurodegeneration and Neuroinflammation Research Group, Girona Biomedical Research Institute (IDIBGI), Dr. Josep Trueta University Hospital, 17001 Catalonia, Spain; 7Department of Medical Sciences, School of Medicine, University of Girona, 17001 Girona, Spain; 8Neuroimmunology and Multiple Sclerosis Unit, Laboratory of Advanced Imaging in Neuroimmunological Diseases, Hospital Clinic Barcelona, Institut d’Investigacions Biomediques August Pi i Sunyer (IDIBAPS) and Universitat de Barcelona, 08036 Barcelona, Spain; 9Servei de Neurologia, Centre d’Esclerosi Múltiple de Catalunya (Cemcat), Institut de Recerca Vall d’Hebrón (VHIR), Hospital Universitari Vall d’Hebrón, Universitat Autònoma de Barcelona, 08907 Barcelona, Spain

**Keywords:** multiple sclerosis, neurofilament light chain, glial fibrillary acidic protein, intrathecal IgM synthesis, progression

## Abstract

The combined use of serum and CSF biomarkers for prognostic stratification in multiple sclerosis (MS) remains underexplored. This multicenter observational study investigated associations between serum neurofilament light chain (sNfL), glial fibrillary acidic protein (sGFAP), and CSF lipid-specific IgM oligoclonal bands (LS-OCMB) with different forms of disability worsening, such as relapse-associated worsening (RAW), active progression independent of relapse activity (aPIRA), and non-active PIRA (naPIRA). A total of 535 patients with MS were included, all sampled within one year of disease onset. Biomarkers were quantified using single-molecule array and immunoblotting techniques, and CSF cell subsets were analyzed by flow cytometry. High sNfL z-scores and LS-OCMB positivity were independently associated with increased risk of RAW and aPIRA, collectively termed inflammatory-associated worsening (IAW), while elevated sGFAP levels predicted naPIRA. Patients with both high sNfL and LS-OCMB positivity had the highest risk of IAW. Among LS-OCMB–positive patients, higher regulatory T cell percentages were associated with lower sNfL levels, suggesting a protective role. Conversely, in LS-OCMB–negative patients, sNfL levels correlated with CSF C3 concentrations. These findings support the complementary role of sNfL, sGFAP, and LS-OCMB in identifying distinct mechanisms of disease worsening and may inform early personalized management strategies in MS.

## 1. Introduction

Multiple sclerosis (MS) management is undergoing a profound transformation and advancing toward a more personalized and precision medicine (PM) approach [1]. The early identification of patients at risk for a more aggressive disease course is crucial for individually tailoring the selection of disease-modifying treatments (DMTs) while minimizing unnecessary exposure to potential risks. Advanced radiological and biological biomarkers are pivotal to enabling this transition to PM [2]. The ultimate therapeutic goal is the prevention of long-term disability, which typically results either from incomplete recovery after relapses—termed relapse-associated worsening (RAW)—or from insidious progression independent of relapse activity (PIRA) [3]. Current DMTs have demonstrated efficacy in reducing relapse rates and mitigating RAW [4], but their success in suppressing overt inflammatory activity has brought to light a neurodegenerative process characterized by non-relapsing progression. This unmasked progression highlights the urgent need for therapies with alternative mechanisms of action and underscores the importance of identifying the immunological pathways driving disease course to enable the effective implementation of PM.

Currently, risk stratification at disease onset for DMT selection relies on demographic, and mainly on clinical, and radiological factors [1]. However, these factors often show limited predictive power and/or require time to accurately predict disease course when neurological damage could already have occurred [4]. Moreover, most clinical and radiological factors do not identify the mechanisms underlying disease worsening. In contrast, biological biomarkers provide a quantitative means of detecting neurological damage at earlier stages [5].

Among the most widely recognized biomarkers are those derived from cerebrospinal fluid (CSF), such as intrathecal IgM synthesis (ITMS) [6,7,8,9], and serum, notably neurofilament light chain (sNfL) [10] and glial fibrillary acidic protein (sGFAP) [11]. ITMS, particularly when detected as lipid-specific IgM oligoclonal bands (LS-OCMB) [9], has been associated with a higher risk of aggressive MS [6,7,8,9]. sNfL levels have been consistently linked to an increased risk of inflammatory-driven worsening [10,12], with high-efficacy DMTs (HE-DMTs) presenting the potential to mitigate this risk [12]. Conversely, sGFAP levels have been associated with non-inflammatory progression in either progressive MS [11] or relapsing-remitting MS (RRMS) [13].

Despite these advances, the interplay between CSF and serum biomarkers in predicting the two primary pathways of disability worsening—RAW and PIRA—remains underexplored. Moreover, PIRA is defined by the absence of overt clinical relapses, but this definition does not account for subclinical inflammatory activity, such as the presence of new MRI lesions. This limits the utility of PIRA as a purely non-inflammatory construct and calls for stratification into active versus non-active PIRA based on MRI findings, which is a distinction rarely considered in prior biomarker research.

The aim of this study was to evaluate the independent value of CSF (LS-OCMB) and serum (sNfL and sGFAP) biomarkers obtained at disease onset for predicting different types of disability worsening (RAW, active PIRA, and non-active PIRA) in a cohort of patients with MS (PwMS). By stratifying PIRA based on the presence or absence of new MRI lesions, we aimed to disentangle the inflammatory and non-inflammatory contributions to progression. Furthermore, we sought to identify distinct immunological signatures associated with these biomarkers by analyzing soluble factors and lymphocyte subsets in the CSF.

## 2. Results

### 2.1. Baseline Characteristics

Six-hundred and two patients were initially selected for this study. We excluded participants with a final diagnosis other than relapsing MS (n = 31) and those who received corticosteroids within two months before sample collection (n = 24) or any DMT prior to sampling (n = 12). After exclusions, 535 patients were included in the analysis. Table 1 provides a summary of their baseline characteristics. Briefly, the median age at sampling was 34.0 years (interquartile range [IQR], 27.5–42.5), and 372 (69.5%) were female. The median follow-up duration was 7.05 years (IQR, 4.93–10.5).

### 2.2. Risk of RAW, Active, and Non-Active PIRA

The 10-year cumulative incidences of RAW, aPIRA, and naPIRA in the total cohort were 21.0%, 15.2%, and 17.1%, respectively. The multivariable Cox regression results indicated that high sNfL levels and the presence of LS-OCMB were independently associated with an increased risk of RAW (sNfL: HR, 2.12; 95% CI, 1.27–3.54; *p* = 0.004; LS-OCMB: HR, 2.15; 95% CI, 1.34–3.45; *p* = 0.002). Similarly, a higher risk of aPIRA was associated with elevated sNfL values (HR, 2.12; 95% CI, 1.17–3.86; *p* = 0.01). Both outcomes were also associated with a higher T2 lesion load and lower proportions of DMT use. On the other hand, only high sGFAP values were associated with an increased risk of naPIRA (HR, 3.19; 95% CI, 1.84–5.34; *p* < 0.001), which was also linked to older age, male sex, and higher baseline EDSS scores (Table 2).

### 2.3. Risk of IAW and naPIRA Across Different Biomarker Groups

Our findings identified sNfL levels, LS-OCMB, and sGFAP concentrations as the most important predictors of disability worsening. Interestingly, the factors associated with RAW and aPIRA were remarkably similar but differed substantially from those linked to naPIRA (Table 2). Consequently, we combined RAW and aPIRA into a single outcome termed inflammatory-associated worsening (IAW).

To further explore the association of biomarker profiles with IAW and naPIRA, we analyzed all possible combinations of the three biomarkers to assess their predictive value for these outcomes. Patients were categorized into the following groups based on serum levels of sNfL and sGFAP and the presence of LS-OCMB in CSF:**Triple Negative (NLGLM−)**: Low sNfL and sGFAP levels and absence of LS-OCMB.**NLGLM+**: Low sNfL and sGFAP levels, and the presence of LS-OCMB.**NLGHM−**: Low sNfL, high sGFAP, and absence of LS-OCMB.**NLGHM+**: Low sNfL, high sGFAP, and the presence of LS-OCMB.**NHGLM−**: High sNfL, low sGFAP, and absence of LS-OCMB.**NHGHM−**: High sNfL and sGFAP levels and absence of LS-OCMB.**NHGLM+**: High sNfL, low sGFAP, and the presence of LS-OCMB.**Triple Positive (NHGHM+)**: High sNfL and sGFAP levels, with the presence of LS-OCMB.

High sNfL levels and/or LS-OCMB positivity were strongly associated with an increased risk of IAW, with the combination of both markers amplifying this risk remarkably. For naPIRA, elevated sGFAP levels and, to a lesser extent, high sNfL were predictive of increased risk, while LS-OCMB positivity alone did not affect the risk of naPIRA. Multivariable Cox regression analyses illustrating the risks of IAW and naPIRA across these groups are shown in Figure 1.

### 2.4. CSF Cells and Soluble Factors Associated with Biomarkers

We next explored potential differences in CSF cells and soluble factors associated with biomarkers related to both outcomes, and they are shown as follows: (1) sNfL and LS-OCMB with the risk of IAW, and (2) sGFAP with the risk of naPIRA. First, we categorized patients into the following four groups based on sNfL levels and LS-OCMB status:sNfL low, LS-OCMB negative (NLM−).sNfL low, LS-OCMB positive (NLM+).sNfL high, LS-OCMB negative (NHM−).sNfL high, LS-OCMB positive (NHM+).

Patients were stratified based on their levels of sGFAP (high vs. low). This analysis was conducted in a subgroup of patients for whom CSF flow cytometry data were available (n = 122). We compared the percentages of CD3+, CD4+, CD8+, CD19+, and monocytes along with their respective subpopulations (Table 3).

The most prominent differences in immune cell subsets were observed among combinations of sNfL and LS-OCMB. These differences were most notable in the CD4+ regulatory T cell (T-reg) population (Table 3). T-reg frequencies were significantly higher in the NLM+ cohort compared to all other groups (Figure 2A). Interestingly, T-reg levels appeared to modulate sNfL concentrations exclusively in patients with positive LS-OCMB, with an area AUC of 0.89 (95% CI 0.76–1.00, *p* < 0.001) (Figure 2B). This finding suggests a potential protective role of regulatory T cells in individuals with intrathecal IgM synthesis.

In multivariable linear regression analysis, LS-OCMB positivity was independently associated with higher sNfL z-scores (*β* = 1.95, 95% CI 0.67–3.24; *p* = 0.003) (Table 4). Additionally, T-regs were independently associated with sNfL levels among patients with positive LS-OCMB (*β* = −0.36, 95% CI −0.55–[−0.16]; *p* = 0.001). Predictive margins, adjusted for age, sex, and time from symptom onset to sampling, demonstrated that higher T-reg frequencies were linked to lower predicted sNfL levels (Figure 2C).

No other significant cellular differences were observed among the four groups, except for reduced CD4+ naïve T cells and increased CD8+ effector memory T cells in the NHM+ cohort (Table 3).

On the other hand, CSF cells associated with sGFAP were mainly terminally differentiated CD8+, which were lower in patients with high levels of sGFAP. No other significant differences were found between both cohorts (Table 3).

We also assessed soluble factors in CSF across groups. The most striking finding involved complement factor C3. The NLM- cohort exhibited significantly lower C3 levels than all other groups (Figure 3A), while C4 values were only lower than the NHM+ cohort. While T-regs appeared to modulate sNfL levels in patients with positive LS-OCMB, C3 levels were the main factor associated with sNfL elevation in patients with negative LS-OCMB (AUC 0.85, 95% CI 0.74–0.95; *p* < 0.001; Figure 3B). In a multivariable linear regression analysis, each doubling of C3 concentration was associated with an increase of 0.49 in sNfL z-score levels in LS-OCMB-negative patients (*β* = 0.49, 95% CI 0.11–0.87; *p* = 0.01; Table 5; Figure 3C).

Given the differences in complement levels, we also assessed the IgG and IgM indices across groups. No significant differences were observed, except for a higher IgM index in patients with positive LS-OCMB (Table 3).

## 3. Discussion

Multiple sclerosis (MS) is a highly variable disease that is characterized by large differences in the rate and extent of disability accrual across patients [14]. The disease encompasses two main immunopathological mechanisms: inflammation and neurodegeneration [15,16]. Inflammation is primarily driven by systemic adaptive immunity with the peripheral activation of B- and T-cells that disrupt the blood–brain barrier and migrate into the central nervous system (CNS) [16,17]. Neurodegeneration, on the other hand, is linked to CNS-compartmentalized inflammation, in which, innate immunity plays a critical role through various mechanisms, such as meningeal follicles, chronic active lesions, and diffuse white matter inflammation [17,18,19] in addition to pure neurodegenerative processes, including Wallerian degeneration, energy deficits, iron and glutamate neurotoxicity, and oxidative stress [20]. These processes contribute to disability worsening in PwMS, primarily through incomplete recovery from relapses (relapse-associated worsening [RAW]) and progression independent of relapse activity (PIRA) [3].

While RAW is associated with acute inflammation, PIRA reflects progression without clinically concurrent and evident relapses and does not distinguish the underlying mechanisms that may involve multiple pathophysiological processes. To refine the identification of mechanisms driving PIRA events, MRI is used to assess recent radiological activity, classifying PIRA as active (aPIRA) or non-active (naPIRA) [21,22]. However, MRI may not detect acute inflammation in the spinal cord (routinely not assessed) or in patients with extensive T2 lesion loads. Consequently, biological biomarkers, especially those detected in the CSF, aim to detect pathophysiological mechanisms more accurately and at earlier stages, thus contributing to early stratification in PwMS [5]. Among these, ITMS, particularly when measured as LS-OCMB, has shown strong and consistent associations with worse disease courses in MS [6,7,8,9,23].

In recent years, serum biomarkers have gained prominence in the field of MS. Serum NfL levels have been consistently associated with disease activity [10,24] and an increased risk of disability worsening [10], including RAW [12,13] and PIRA [12,13,24]. Conversely, sGFAP levels have been specifically linked to PIRA, particularly in cases with low sNfL levels [11,13]. As biomarker research becomes increasingly complex, a deeper understanding of the independent and combined roles of these biomarkers is critical. This knowledge will not only help elucidate underlying immunopathological mechanisms but also facilitate risk stratification and inform optimized treatment selection.

Our study offers several important insights. First, we identified factors that were shared by patients presenting RAW and/or aPIRA, including T2 lesion load, sNfL levels, and LS-OCMB. Based on these similarities, we proposed a unifying concept, namely IAW, to incorporate both outcomes. This approach allowed us to delineate two distinct pathways of disability worsening, IAW and naPIRA. Importantly, we observed that IAW could be mitigated by DMTs, which is a benefit not observed with naPIRA.

Second, we confirmed the role of sGFAP as a biomarker of chronic inflammation and naPIRA, which is consistent with prior studies [11,13]. Previous research has shown that sGFAP is associated with PIRA primarily in the context of low inflammation, which underscores its capability to capture chronic, rather than acute, inflammation. In our study, sGFAP concentrations were specifically linked to naPIRA, further supporting its capability to identify patients at risk of “true” PIRA [25].

Next, we observed that both sNfL and LS-OCMB were independently associated with an increased risk of IAW. A synergistic effect was evident when both biomarkers were positive, which nearly doubled the risk compared to patients with only one positive biomarker. In support of this finding, CSF NfL levels were higher in the NHM+ cohort even when compared to the NHM- group, a finding that reinforced the evidence indicating greater axonal damage in this cohort.

Finally, we explored the potential mechanisms contributing to the poorer prognosis observed in each group by analyzing CSF cell subsets and soluble factors. We found that increasing C3 levels were strongly associated with elevated sNfL z-score values, particularly in patients with negative LS-OCMB. In contrast, higher CSF T-regs were linked to lower sNfL levels, especially in patients with positive LS-OCMB.

Complement factors have gained considerable attention in recent years due to their association with poor prognoses, particularly in patients with ITMS. Increased complement levels have been linked to higher EDSS and MSSS scores, as well as elevated sNfL values [26,27]. Indeed, the aggressive disease course observed in patients with ITMS may be partly explained by complement activation and the increased presence of autoreactive B cells in the CSF (particularly the CD5+ subsets) [8,28].

In our study, we observed that complement factors failed to distinguish between high and low sNfL levels if a patient was LS-OCMB-positive. Conversely, in patients with negative LS-OCMB, higher C3 levels were strongly linked to increased sNfL concentrations even in the context of similar IgG or IgM index values. Previous reports have described the colocalization of IgG and C3b in MS lesions [29] and molecular mimicry between the antigen EBNA1 from Epstein Barr virus and antibodies directed to proteins in PwMS [30]. However, more evidence is needed to decide whether antibodies against any particular antigen may be linked to high C3 and sNfL levels in patients with negative LS-OCMB.

On the other hand, we observed that higher CSF T-reg percentages were associated with lower sNfL levels among patients with positive LS-OCMB, suggesting that the poorer prognosis linked to this group might be partially mitigated by the presence of T-regs.

The role of regulatory T-cells in MS is well-established, as these cells maintain the peripheral tolerance and regulate autoreactive immune responses [31]. Numerous studies demonstrated that CD4+CD25+FoxP3+ T-regs are reduced in PwMS compared to healthy controls [32], with further reductions observed during relapses [33]. In this line, lower T-reg levels are associated with worse clinical and radiological outcomes [34].

In our study, T-regs appeared to mitigate axonal damage and inflammation in patients with positive LS-OCMB, despite the elevated complement factors observed in these patients. This finding is particularly compelling, as IgM deposits have been found in MS lesions with activated complement factor C3b [29], a key driver of axonal damage. These results underscore the protective role of T-regs in mitigating axonal damage and inflammation by counteracting critical immunological effector mechanisms and emphasize their potential as promising therapeutic tools.

In contrast, patients with high sGFAP levels showed few significant differences, aside from a reduction in CD8+ terminally differentiated cells. These findings suggest that naPIRA—typically occurring at more advanced disease stages—may be associated with later alterations in CNS immune cell profiles or with distinct mechanisms involving CNS-resident microglia [16,17,18].

Our findings suggest a biomarker-driven approach to selecting a treatment option. “Triple negative” patients (NLGLM-) are at low risk of inflammatory outcomes and may benefit from a broad range of DMTs based on other clinical factors. The presence of LS-OCMB or high sNfL levels, independent of each other, should prompt the consideration of HE-DMTs, and early use is warranted especially if both biomarkers are present [12,13]. These patients are also at risk of later naPIRA, a finding that likely suggests that insufficient control of early peripheral inflammation might contribute to subsequent CNS-compartmentalized chronic inflammation. Last, patients with high sGFAP levels are at elevated risk of naPIRA and may benefit from emerging therapies targeting chronic inflammation [35]. This biomarker-driven approach has the potential to improve outcomes by personalizing treatment strategies according to specific immunopathological processes.

The present study has several limitations. First, flow cytometry analyses were only conducted in a subset of the total cohort. This could limit the statistical power to detect significant differences between cohorts and requires further validation in subsequent studies. For instance, the correlation between T-reg frequencies and sNfL levels was significant only in LS-OCMB-positive patients, which is likely due to the higher inflammatory profile with elevated sNfL levels in this subgroup. Further studies are needed to clarify the relationship between T-reg cells, sNfL levels, and ITMS. Second, the classification of patients into aPIRA and naPIRA relied exclusively on brain MRI, as spinal cord assessments in temporal proximity to PIRA events were exceedingly rare. This reliance on MRIs may have led to an underestimation of aPIRA cases, potentially skewing results toward naPIRA. Nevertheless, the strikingly similar findings between RAW and aPIRA suggest that any underestimation of aPIRA was likely minimal and did not affect the overall conclusions.

In summary, our study highlights the utility of integrating biomarkers to dissect the complex pathophysiological mechanisms underlying disability worsening in MS. These results underscore the potential for biomarker-driven stratification to guide early and personalized treatment decisions. Further research, including longitudinal studies and randomized clinical trials, is warranted to validate these findings and integrate them into clinical practice, ultimately enabling a precise clinical approach when treating PwMS.

## 4. Materials and Methods

### 4.1. Study Design

This multicenter observational study with prospective data collection was conducted at Hospital Universitario Ramón y Cajal (Madrid, Spain). The study adhered to the Strengthening the Reporting of Observational Studies in Epidemiology (STROBE) guidelines. We recruited PwMS from seven university hospitals (Table A1) who presented with a first demyelinating relapse (in the absence of any clinical manifestations) and had stored serum and CSF samples obtained within 12 months after disease onset and a baseline magnetic resonance imaging (MRI) conducted within six months of the initial relapse. Exclusion criteria included treatment with DMTs before sample collection, use of corticosteroids within two months prior to sampling, and any diagnosis other than relapsing–remitting MS.

### 4.2. Data Collection

We included all consecutive patients recruited between 15 July 1994, and 22 April 2023, who fulfilled the inclusion criteria and underwent the last follow-up conducted on 11 October 2024. Baseline data collection included demographic, clinical, and radiological variables. We recorded the dates of initiation and discontinuation for all administered drugs, which were classified into the following two groups: (1) injectable/oral DMTs (glatiramer acetate, all interferon-β formulations, fumarates, teriflunomide, sphingosine-1-phosphate receptor modulators, cladribine, or azathioprine) and (2) monoclonal antibodies ([mAbs], including alemtuzumab, natalizumab, ocrelizumab, rituximab, and ofatumumab).

To estimate the type and duration of treatments, we calculated the proportion of time each patient spent in each DMT group by dividing the duration of each treatment by the total time elapsed until the respective outcome of interest. The primary outcomes were time to the first RAW and/or active/non-active PIRA event. All Expanded Disability Status Scale (EDSS) assessments were evaluated by experienced neurologists at a minimum of every six months with additional evaluations in case of relapses.

### 4.3. CSF Analyses

Blood and CSF samples were collected concurrently. Serum samples were aliquoted and stored at –80 °C until used for analysis. CSF samples (6–8 mL) were centrifuged at 500× *g* for 15 min, after which the cellular pellets were processed. Additionally, CSF was aliquoted and stored at –80 °C until assayed. Serum and CSF IgG, IgM, and albumin concentrations were measured using a BN ProSpec nephelometer (Siemens Healthcare Diagnostics, Marburg, Germany). OCMB and lipid-specific OCMB (LS-OCMB) were analyzed in serum and CSF through isoelectric focusing and immunoblotting as previously described [8].

Monoclonal antibodies used for the analysis of CSF cells included CD3 PerCP, CD5 APC, CD8 APC-H7, CD14 FITC, CD19 PE-Cy7, CD24 PE, CD25 PE, CD27 FITC, CD38 PE-Cy5, CD45 V500, CD45RO APC, CD127 BV421, and CD197 (CCR7) PE (all from BD Biosciences, San Jose, CA, USA) (Table A2). We followed the gating strategy as shown in Figure A1.

The cell pellets obtained after CSF centrifugation were resuspended in the residual volume and incubated in the dark at 4 °C with appropriate concentrations of monoclonal antibodies for 30 min. Samples were subsequently washed twice with phosphate-buffered saline (PBS), and cell counts were acquired using a fluorescence-activated cell sorting (FACS) Canto II flow cytometer (BD Biosciences, San Jose, CA, USA). The percentages of different CSF cell subpopulations were analyzed using FACSDiva software version 8.0 (BD Biosciences, San Jose, CA, USA).

### 4.4. Soluble Factors

sNfL and sGFAP levels were quantified using the SR-X instrument with the Simoa^®^ NF-Light™ Advantage Kit and the Simoa^®^ GFAP™ Discovery Kit (Quanterix, Billerica, MA, USA), respectively, following the manufacturer’s protocols. Complement C3 and C4 levels in CSF were measured using the Human Complement C3 and C4 enzyme-linked immunosorbent assay (ELISA) Kits (Abcam, Cambridge, UK), respectively, in accordance with the manufacturer’s instructions.

### 4.5. MRI Protocols

MRI was performed at 1.5 T magnet (Ingenia or Achieva; Philips Healthcare, Best, The Netherlands) with a slide thickness of 2–5 mm. The protocols of image acquisition included the following pulse sequences in the axial plane: T1: weighted spin-echo; T2: weighted turbo spin-echo, proton density-weighted turbo spin-echo, and/or fluid-attenuated inversion recovery sequences. Spinal cord studies included sagittal T1 and T2 weighted with either PD or short tau inversion recovery as well as guided axial T2-weighted sequences.

### 4.6. Definitions

The 2017 revised McDonald criteria were used for the diagnosis of MS. Definitions of RAW and PIRA were derived from the literature [3,36]. A standardized definition of PIRA that incorporated a roving baseline, a stepwise, stratified increase in EDSS, and confirmation of progression at least six months later was applied [36]. PIRA was subclassified as active (aPIRA) or non-active (naPIRA) based on the presence or absence of new T2 and/or contrast-enhancing lesions within one year of the PIRA event.

sNfL levels were adjusted for age and body mass index (BMI) using a standardized z-score [10]. A z-score of 1.5 was applied as the cut-off to define high sNfL levels based on the established literature [10,12,13]. The cut-offs for sGFAP levels were set at 140 pg/mL for patients younger than 55 years and 280 pg/mL for those aged 55 or older based on references from a recent cohort of healthy donors in our center [37].

### 4.7. Statistical Analyses

For descriptive analyses, categorical variables were expressed as absolute and relative proportions, with differences assessed using χ^2^ or Fisher’s exact test. Continuous variables were described using the median and interquartile range (IQR), and group associations were evaluated using an analysis of variance (ANOVA) with Bonferroni correction for multiple comparisons. Multivariable Cox proportional hazard regression models were employed to examine the associations between CSF and serum biomarkers and the risk of RAW, aPIRA, and naPIRA. Risk estimates were reported as hazard ratios (HRs) with 95% confidence intervals (CIs).

The multivariable models were adjusted for the following covariates: (1) age at disease onset, (2) sex, (3) baseline EDSS, (4) T2 lesion load, and (5) the proportion of time patients were treated with either mAbs or injectable/oral DMTs. Harrell’s C index was used as a measure of goodness of fit, and the proportional hazard assumption was tested using a regression of the scaled Schoenfeld residuals against time. Multivariable linear regressions were used to assess associations between CSF cells or soluble factors and sNfL levels. Quantitative variables with skewed distributions (C3 levels) were log-transformed, and the estimates were back-transformed to represent multiplicative effects. Receiver operating characteristic (ROC) curves and the Youden index were applied to determine area under the curve (AUC) and cut-off values.

All analyses were conducted using Stata, version 17 (StataCorp LLC, College Station, TX, USA), and GraphPad Prism 9.5 software (GraphPad Prism Inc., La Jolla, CA, USA). A *p*-value of <0.05 was considered statistically significant, and all tests were two-tailed.

## Figures and Tables

**Figure 1 ijms-26-06898-f001:**
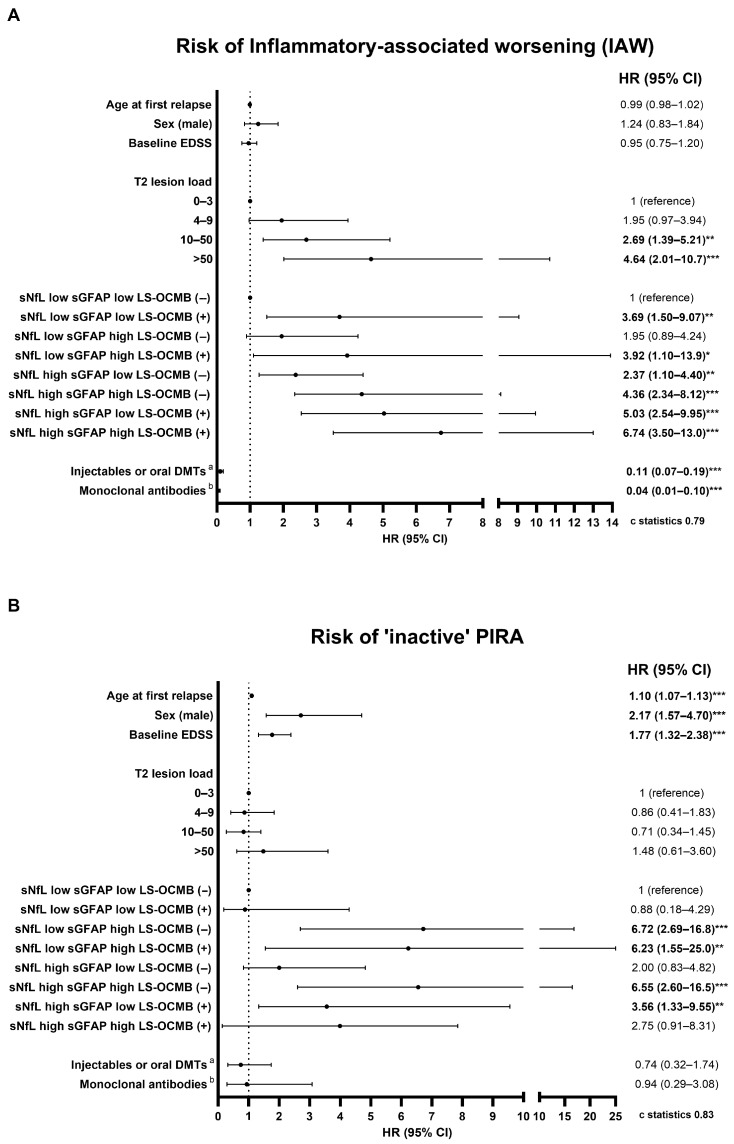
Multivariable Cox regressions of the risk of IAW and non-active PIRA. Estimation of the risk of IAW (**A**) and non-active PIRA (**B**) in patients categorized by CSF and serum biomarkers status. Results are presented as adjusted hazard ratios (HRs) with 95% confidence intervals (CIs). ^a^ Injectable/oral DMTs: glatiramer acetate, all interferon-β formulations, fumarates, teriflunomide, sphingosine-1-phosphate receptor modulators, cladribine, or azathioprine. ^b^ Monoclonal antibodies: natalizumab, alemtuzumab, ocrelizumab, rituximab, ofatumumab. * *p* < 0.05; ** *p* < 0.001; *** *p* < 0.001.

**Figure 2 ijms-26-06898-f002:**
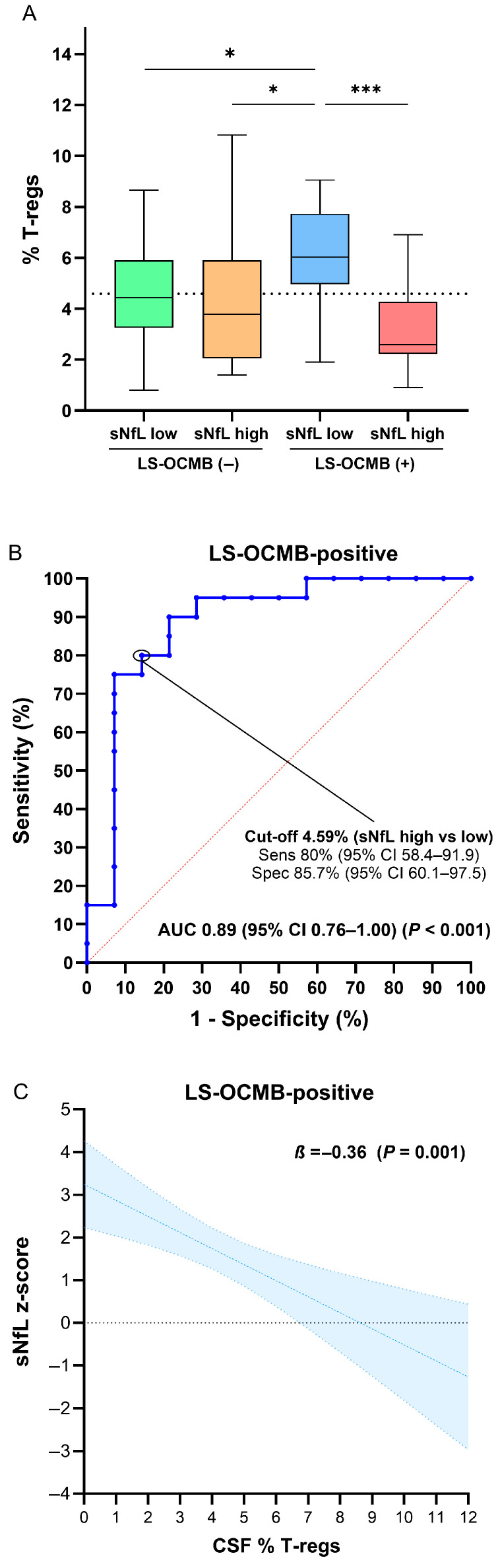
Differences in CSF regulatory T cells based on LS-OCMB and sNfL status. Box plots illustrate differences in regulatory T cell (T-reg) percentages across four groups categorized by the presence or absence of LS-OCMB and high or low sNfL concentrations (**A**). The ROC curve shows that T-reg cells effectively distinguished between high and low sNfL levels in patients with positive LS-OCMB (**B**). Predictive margins derived from multivariable linear regressions (adjusted by age, sex, and time from symptom onset to sampling) demonstrate that higher T-reg percentages were associated with decreasing sNfL z-score values in patients with positive LS-OCMB (**C**). * *p* < 0.05; *** *p* < 0.001.

**Figure 3 ijms-26-06898-f003:**
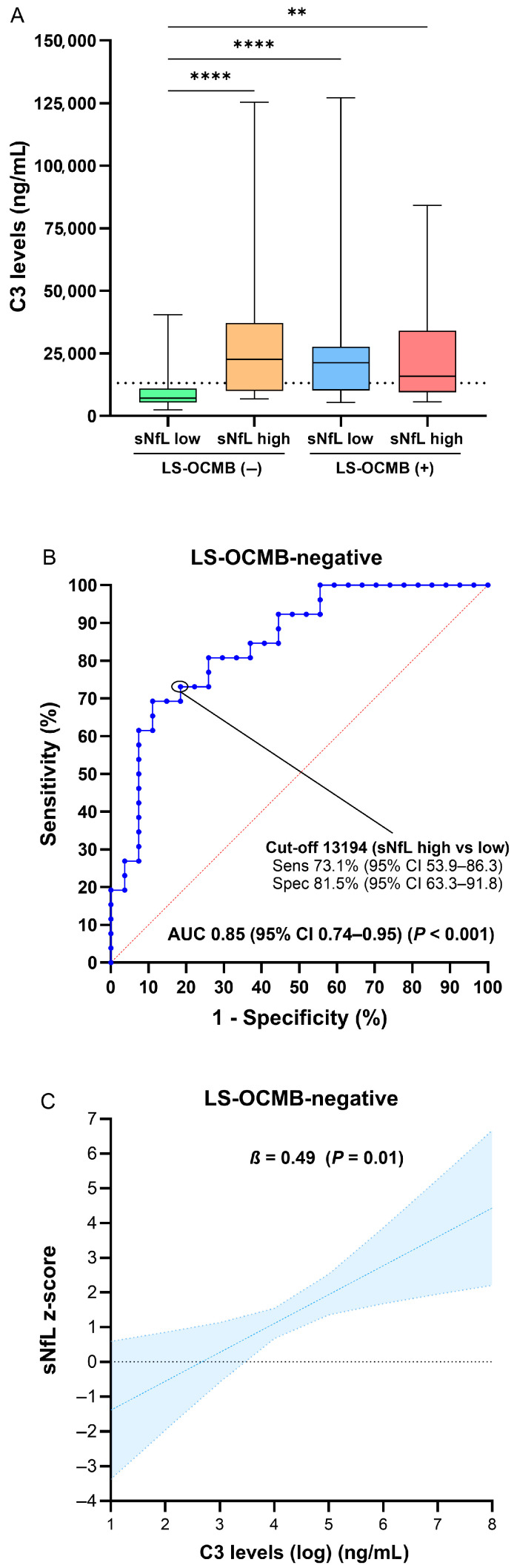
Differences in CSF complement C3 levels based on LS-OCMB and sNfL status. Box plots show levels in complement C3 across four groups categorized by the presence or absence of LS-OCMB and high or low sNfL levels (**A**). C3 levels show potential to discriminate patients with high sNfL z-score values if LS-OCMB is negative (**B**). Predictive margins for C3 levels were estimated from multivariable linear regressions and indicate a correlation with sNfL z-scores in patients with negative LS-OCMB (**C**). ** *p* < 0.01; **** *p* < 0.0001.

**Table 1 ijms-26-06898-t001:** Patient characteristics.

	Total (n = 535)
Sex (female)	372 (69.5)
Age at first symptom, y	33.8 (27.2–42.2)
Age at serum analysis, y	34.0 (27.5–42.5)
Time to analysis after first relapse, mo	3.12 (0.59–6.23)
Topography of first relapse	
Optic nerve	96 (17.9)
Brainstem	124 (23.2)
Spinal cord	219 (40.9)
Cerebral hemisphere	59 (11.0)
Multifocal	26 (4.9)
Paroxysmal symptoms	11 (2.1)
EDSS at baseline	1.5 (1–2)
T2 lesions at baseline	
0–3	84 (15.7)
4–9	142 (26.5)
10–50	258 (48.2)
>50	51 (9.5)
Gadolinium-enhancing lesions	
Median (range)	1 (0–45)
No. of patients with enhancing lesions (%)	275/486 (56.6)
CSF data	
IgG oligoclonal bands	489 (91.4)
IgM oligoclonal bands	259 (48.4)
Lipid-specific IgM oligoclonal bands	170 (31.8)
Serum biomarkers levels	
sNfL levels (pg/mL)	11.1 (6.84–19.7)
sNfL *z*-score	1.68 (0.44–2.58)
sGFAP levels (pg/mL)	126.5 (90.9–181.5)
DMT use during follow-up ^1^	
None	89 (16.6)
Injectable/oral DMTs ^2^	380 (71.0)
Monoclonal antibodies ^3^	166 (31.0)
Time of follow-up, y	7.05 (4.93–10.5)
Patients attaining 6-month CDW during follow-up ^4^	
RAW	86 (16.1)
‘Active’ PIRA	61 (11.4)
‘Non-active’ PIRA	72 (13.5)

Categorical variables are shown as a number (%). Continuous variables are described as a median (interquartile range). Categorical variables are shown as a number (%). ^1^ Some patients received both groups of treatments. ^2^ Injectable/oral DMTs: glatiramer acetate, all interferon-β formulations, fumarates, teriflunomide, sphingosine-1-phosphate receptor modulators, cladribine, or azathioprine. ^3^ Monoclonal antibodies: natalizumab, alemtuzumab, ocrelizumab, rituximab, ofatumumab. ^4^ Some patients might have experienced RAW, active PIRA, and/or inactive PIRA during their follow-up.

**Table 2 ijms-26-06898-t002:** Multivariable Cox regression models assessing the risk of RAW, aPIRA, and naPIRA.

Variables	RAW		Active PIRA		Non-Active PIRA	
	HR (95% CI)	*p*	HR (95% CI)	*p*	HR (95% CI)	*p*
	c Statistics 0.82		c Statistics 0.74		c Statistics 0.82	
**Age at first relapse**	0.98 (0.96–1.00)	0.05	1.03 (0.99–1.05)	0.05	**1.10 (1.07–1.13)**	**<0.001**
**Sex (male)**	1.45 (0.91–2.32)	0.12	1.26 (0.70–2.27)	0.44	**2.87 (1.69–4.88)**	**<0.001**
**Baseline EDSS**	1.09 (0.82–1.44)	0.55	0.71 (0.51–1.01)	0.05	**1.71 (1.30–2.25)**	**<0.001**
**T2 lesion load**						
**0–3 (reference)**	1	-	1	-	1	-
**4–9**	1.93 (0.87–4.26)	0.11	3.14 (0.90–10.9)	0.07	1.03 (0.49–2.15)	0.95
**10–50**	**2.30 (1.08–4.88)**	**0.03**	**3.48 (1.04–11.7)**	**0.04**	0.63 (0.31–1.28)	0.20
**>50**	**3.98 (1.47–10.7)**	**0.006**	**4.98 (1.23–20.2)**	**0.02**	1.23 (0.52–2.91)	0.64
**sNfL z-score >1.5**	**2.12 (1.27–3.54)**	**0.004**	**2.12 (1.17–3.86)**	**0.01**	1.44 (0.84–2.46)	0.18
**High sGFAP levels ^1^**	1.52 (0.97–2.40)	0.07	1.51 (0.87–2.62)	0.15	**3.19 (1.84–5.34)**	**<0.001**
**Lipid-specific IgM OCB**	**2.15 (1.34–3.45)**	**0.002**	1.70 (0.96–3.02)	0.07	0.82 (0.46–1.47)	0.51
**Proportion of time of injectable/oral DMTs use ^2^**	**0.06 (0.03–0.11)**	**<0.001**	**0.38 (0.17–0.87)**	**0.02**	0.86 (0.38–1.96)	0.72
**Proportion of time of mAb use ^3^**	**0.02 (0.01–0.07)**	**<0.001**	**0.09 (0.02–0.39)**	**0.001**	0.85 (0.26–2.73)	0.78

Multivariable Cox regression models include demographical (age at disease onset and sex), clinical (baseline EDSS), radiological (T2 lesion load) attributes and biomarkers (sNfL and sGFAP levels, and LS-OCMB), including the proportion of time patients were treated with either mAbs or injectable/oral DMTs. Harrell’s C index was used as a measure of goodness of fit. ^1^ sGFAP high: >140 pg/mL for patients younger than 55 years and >280 pg/mL for patients aged 55 or older. ^2^ Injectables or oral DMTs: glatiramer acetate, all interferon-β formulations, fumarates, teriflunomide, sphingosine-1-phosphate receptor modulators, cladribine, or azathioprine. ^3^ mAb: natalizumab, alemtuzumab, ocrelizumab, rituximab, ofatumumab. Bold text highlights significant results.

**Table 3 ijms-26-06898-t003:** Percentage of CSF cell numbers and soluble factors between the different biomarker groups.

	sNfL Low LS-OCMB (−) (n = 45)	sNfL Low LS-OCMB (+) (n = 16)	sNfL High LS-OCMB (−) (n = 36)	sNfL High LS-OCMB (+) (n = 25)	sGFAP Low (n = 75)	sGFAP High (n = 47)
**Lymphocyte subset**						
CD3+ T cells	89.8 (86.8–91.8)	90.8 (87.3–92.6)	89.5 (87.6–90.7)	87.2 (82.4–91.1)	89.5 (86.6–91.2)	89.1 (86.8–91.5)
CD4+	68.8 (61.8–74.4)	67.6 (65.8–71.6)	67.8 (63.0–73.6)	62.3 (56.4–67.1)	67.8 (61.5–71.9)	66.8 (60.2–72.1)
Naive	**6.51 (5.20–8.48)**	**7.22 (5.75–11.4)**	**6.77 (4.23–8.80)**	**1.11 (0.72–2.44) ^1^**	5.77 (4.23–8.18)	6.77 (5.16–8.84)
Central memory	16.5 (13.6–26.4)	30.3 (19.0–31.9)	17.6 (11.9–27.6)	11.2 (9.68–15.7)	16.5 (9.90–26.6)	19.9 (14.7–29.4)
Effector memory	27.8 (24.6–37.7)	26.6 (17.9–41.3)	27.4 (22.1–34.0)	40.2 (37.2–46.4)	27.1 (24.7–37.7)	27.4 (22.1–40.8)
Regulatory	**4.35 (3.20–5.80)**	**5.69 (4.60–7.59) ^1^**	**3.98 (2.07–5.68)**	**2.7 (2.25–4.38)**	4.33 (2.60–5.49)	3.78 (2.25–6.12)
Terminally differentiated	10.2 (5.63–17.9)	7.27 (3.01–9.21)	7.60 (6.14–15.3)	5.40 (2.75–6.34)	9.21 (4.50–15.3)	6.27 (3.20–7.60)
CD8+	18.5 (15.5–21.6)	19.3 (16.0–22.5)	18.7 (15.8–23.5)	20.3 (16.0–24.6)	19.3 (16.1–23.0)	18.7 (15.4–23.5)
Naive	1.15 (0.82–2.09)	2.02 (0.66–2.72)	1.31 (1.18–2.68)	1.18 (0.38–1.94)	1.74 (0.80–2.42)	1.30 (0.66–2.72)
Central memory	0.62 (0.41–1.10)	1.36 (0.86–4.17)	1.03 (0.62–1.36)	0.74 (0.53–1.50)	0.84 (0.50–1.19)	1.31 (0.62–4.17)
Effector memory	**6.57 (4.21–8.13) ^2^**	7.81 (6.93–9.82)	**6.38 (3.14–9.27) ^2^**	**14.1 (10.2–16.0)**	6.61 (4.14–8.07)	9.27 (6.38–10.7)
Terminally differentiated	7.84 (6.03–11.7)	5.81 (5.43–9.23)	10.0 (3.65–12.3)	7.99 (7.67–11.1)	**9.04 (6.84–13.0)**	**5.69 (3.65–9.68) ^4^**
CD19+ B cells	2.60 (1.95–4.30)	3.63 (2.56–4.43)	2.37 (1.45–4.00)	4.25 (2.01–5.70)	2.92 (1.91–4.43)	2.40 (1.80–4.67)
CD5+	0.48 (0.20–0.80)	0.60 (0.20–0.77)	0.40 (0.20–0.75)	0.85 (0.30–1.00)	0.52 (0.20–0.90)	0.44 (0.21–0.90)
CD5-	2.10 (1.70–2.63)	2.66 (1.80–3.28)	1.70 (1.20–3.10)	3.05 (1.67–4.70)	2.17 (1.60–3.30)	2.00 (1.30–3.90)
Memory	1.80 (1.25–2.69)	2.84 (2.38–4.41)	1.65 (1.07–2.44)	2.19 (1.50–4.96)	2.00 (1.41–2.79)	1.50 (1.04–2.80)
Plasmablasts	0.43 (0.32–0.56)	0.55 (0.26–0.70)	0.42 (0.24–1.03)	0.28 (0.10–0.48)	0.48 (0.28–0.80)	0.26 (0.19–0.59)
Monocytes	2.71 (1.46–4.29)	1.61 (1.30–2.70)	3.20 (1.07–4.67)	1.35 (0.80–2.78)	2.40 (1.39–3.86)	2.10 (0.93–4.13)
**Soluble factors**						
NfL levels (pg/mL)	**577 (392–883.6)**	**795 (472–1166)**	**2680 (1311–4691) ^1^**	**3758 (1838–7575) ^1^**	**972 (576–2652)**	**2274 (784–5083) ^4^**
C3 levels (ng/mL)	**7131 (5398–10,933) ^1^**	**12,016 (9244–25,129)**	**14,121 (8610–21,003)**	**12,178 (8387–20,800)**	15,101 (9212–25,838)	21,065 (11,385–45,722)
C4 levels (ng/mL)	**899 (730–1107) ^2^**	1189 (891.8–1699)	1096 (826.8–1957)	**1257 (912–1629)**	**2133 (1082–3545)**	**2452 (1980–4594) ^4^**
IgG index	0.81 (0.60–1.34)	0.95 (0.71–1.38)	0.84 (0.63–1.17)	0.84 (0.68–1.32)	0.84 (0.64–1.32)	0.85 (0.64–1.23)
IgM index	**0.14 (0.08–0.23) ^2,3^**	**0.24 (0.15–0.47)**	**0.13 (0.08–0.22) ^2,3^**	**0.24 (0.15–0.42)**	0.15 (0.09–0.28)	0.16 (0.09–0.28)

Group associations were evaluated using an analysis of variance (ANOVA) with Bonferroni correction for multiple comparisons. We analyzed differences across the cohorts defined by the combination of sNfL and LS-OCMB status (columns 2 to 5), as these biomarkers were associated with inflammation-related outcomes (RAW and aPIRA). Additionally, we compared patients with high versus low sGFAP levels (columns 6 and 7), given the association of sGFAP with naPIRA. ^1^ *p* < 0.05 vs. all other groups, ^2^ *p* < 0.05 vs. “sNfL low LS-OCMB (−)”, ^3^ *p* < 0.05 vs. “sNfL high LS-OCMB (−)”, ^4^ *p* < 0.05 vs. “sGFAP low”. Bold text highlights significant results.

**Table 4 ijms-26-06898-t004:** Multivariable linear regression showing associations between CSF T-reg and sNfL z-score levels.

	Estimate (95% CI)	*p* Value
**Total (n = 108)**		
**Intercept**	1.38 (−0.01–2.77)	0.051
Sex (male)	−0.12 (−0.70–0.47)	0.69
Age at sampling, y	0.00 (−0.03–0.03)	0.98
Time to sampling, mo	−0.01 (−0.09–0.06)	0.75
**CSF T-reg cells, %**	−0.05 (−0.21–0.11) *	0.53
**LS-OCMB (+)**	**1.95 (0.67–3.24) ***	**0.003**
**LS-OCMB-positive (n = 39)**		
**Intercept**	**3.38 (1.43–5.33)**	**0.001**
Sex (male)	−0.43 (−1.31–0.44)	0.32
Age at sampling, y	−0.01 (−0.06–0.05)	0.80
Time to sampling, mo	0.04 (−0.06–0.14)	0.43
**CSF T-reg cells, %**	**−0.36 (−0.55**–[**−0.16])**	**0.001**
**LS-OCMB-negative (n = 69)**		
Intercept	1.41 (−0.24–3.06)	0.09
Sex (male)	0.09 (−0.70–0.88)	0.82
Age at sampling, y	0.01 (−0.03–0.04)	0.80
Time to sampling, mo	−0.06 (−0.16–0.05)	0.29
CSF T-reg cells, %	−0.06 (−0.23–0.11)	0.51

* *p* interaction between T-reg cells and LS-OCMB = 0.02. Interpretation: When all other variables are held constant, sNfL z-score levels decrease by 0.36 units for every 1% increase in CSF T-reg percentages in patients with positive LS-OCMB. The intercept represents the predicted sNfL z-score when all explanatory variables are at zero, which reflects a theoretical baseline value under these conditions. No other significant associations were observed in this model. Bold text highlights significant results.

**Table 5 ijms-26-06898-t005:** Multivariable linear regression showing associations between C3 levels and sNfL z-score values.

	Estimate (95% CI)	*p* Value
**Total (n = 109)**		
Intercept	−4.85 (−10.2–[0.49])	0.08
Sex (male)	0.08 (−0.54–0.71)	0.79
Age at sampling, y	−0.01 (−0.04–0.02)	0.57
Time to sampling, mo	−0.05 (−0.14–0.04)	0.27
**C3 levels (ng/mL) (per doubling)**	**0.47 (0.11–0.84) ***	**0.01**
**LS-OCMB (+)**	**8.39 (1.10–15.7) ***	**0.03**
**LS-OCMB-positive (n = 58)**		
Intercept	3.38 (−1.83–8.60)	0.20
Sex (male)	0.06 (−0.78–0.90)	0.89
Age at sampling, y	0.02 (−0.03–0.06)	0.52
Time to sampling, mo	−0.05 (−0.17–0.06)	0.35
C3 levels (ng/mL) (per doubling)	−0.14 (−0.52–0.24)	0.45
**LS-OCMB-negative (n = 51)**		
Intercept	−4.29 (−9.91–1.33)	0.13
Sex (male)	0.13 (−0.85–1.11)	0.79
Age at sampling, y	−0.04 (−0.08–0.01)	0.13
Time to sampling, mo	−0.03 (−0.18–0.13)	0.76
**C3 levels (ng/mL) (per doubling)**	**0.49 (0.11–0.87)**	**0.01**

* *p* interaction between C3 levels and LS-OCMB = 0.03. Estimates from linear models were back transformed and represent multiplicative effects. Interpretation: When all other variables are held constant, sNfL z-score levels increase by 0.47 and 0.49 units for every doubling of C3 levels in total or patients with negative LS-OCMB, respectively. The intercept represents the predicted sNfL z-score when all explanatory variables are at zero, which reflects a theoretical baseline value under these conditions. No other significant associations were observed in this model. Bold text highlights significant results.

## Data Availability

Anonymized data not published within this article will be made available by request from any qualified investigator.

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
