# Peer review of "Combining CSF and Serum Biomarkers to Differentiate Mechanisms of Disability Worsening in Multiple Sclerosis"

_ijms, 2025, doi:10.3390/ijms26146898_

Round 1

Reviewer 1 Report

Comments and Suggestions for Authors

The manuscript titled “Combining CSF and Serum Biomarkers to Differentiate Mechanisms of Disability Worsening in Multiple Sclerosis” investigates potential biomarkers for Multiple Sclerosis (MS). The size of the patient cohort provides valuable data for clinical diagnosis and may aid in improving patient care. Below are several comments and questions that I hope will help further strengthen the manuscript:

  1. The research team has previously published multiple studies on similar or related MS biomarkers. Is there any overlap in patient data between this cohort and those in previously published articles?
  2. What is the novelty or improvement of this study compared to previously published MS biomarker research by the same or other groups, especially the similar articles published by authors’ lab? This would help convey the unique contribution of this work to the research community.
  3. Please specify the statistical methods used for each table. Additionally, are the reported p-values adjusted or unadjusted? Given the number of categories analyzed, was any correction for multiple comparisons applied?
  4. How did the authors define and categorize the high and low levels of sNfL and sGFAP? Please provide criteria or thresholds used for grouping.
  5. The introduction is somewhat insufficient and would benefit from additional background information. In particular, key concepts such as relapse-associated worsening (RAW) and progression independent of relapse activity (PIRA) should be clearly defined and discussed. Explaining why these phenomena are important and why they warrant investigation would help provide better context for the study. Additionally, the authors should consider ending the introduction with a clear statement of the study’s aim and its novelty. This will help readers understand the unique contribution of the work from the outset.

Author Response

Reviewer #1

The manuscript titled “Combining CSF and Serum Biomarkers to Differentiate Mechanisms of Disability Worsening in Multiple Sclerosis” investigates potential biomarkers for Multiple Sclerosis (MS). The size of the patient cohort provides valuable data for clinical diagnosis and may aid in improving patient care. Below are several comments and questions that I hope will help further strengthen the manuscript:

  1. The research team has previously published multiple studies on similar or related MS biomarkers. Is there any overlap in patient data between this cohort and those in previously published articles?

Response: We thank the reviewer for this point. We included patients used in previous studies (PMID: 36848127 and PMID: 39101570) in which we had available CSF samples and CSF IgM OCB data. However, we recruited additional patients for this study.

  1. What is the novelty or improvement of this study compared to previously published MS biomarker research by the same or other groups, especially the similar articles published by authors’ lab? This would help convey the unique contribution of this work to the research community.

Response: We thank the reviewer for this valuable comment. To better highlight the novelty and contribution of our study, we have revised the Introduction section to clarify the unique aspects of our approach. Specifically, we have emphasized the need to distinguish between the different mechanisms underlying disability worsening in MS—namely relapse-associated worsening (RAW) and progression independent of relapse activity (PIRA)—and introduced the novel concept of stratifying PIRA based on the presence or absence of subclinical inflammatory activity (i.e., new MRI lesions). This distinction, rarely addressed in prior biomarker studies, allows for a more nuanced evaluation of the prognostic value of biological markers.

Accordingly, the following modifications have been made in the manuscript (new text in bold):

Page 2, lines 67–76:

Multiple sclerosis (MS) management is undergoing a profound transformation and advancing toward a more personalized and precision medicine (PM) approach [1]. Early identification of patients at risk for a more aggressive disease course is crucial for individually tailoring the selection of disease-modifying treatments (DMTs) while minimizing unnecessary exposure to potential risks. Advanced radiological and biological biomarkers are pivotal to enabling this transition to PM [2]. The ultimate therapeutic goal is the prevention of long-term disability, which typically results either from incomplete recovery after relapses—termed relapse-associated worsening (RAW)—or from insidious progression independent of relapse activity (PIRA) [3]. Current DMTs have demonstrated efficacy in reducing relapse rates and mitigating RAW [4], but their success in suppressing overt inflammatory activity has brought to light a neurodegenerative process characterized by non-relapsing progression. This unmasked progression highlights the urgent need for therapies with alternative mechanisms of action and underscores the importance of identifying the immunological pathways driving disease course to enable effective implementation of PM.

Page 3, lines 93–100:

Despite these advances, the interplay between CSF and serum biomarkers in predicting the two primary pathways of disability worsening—RAW and PIRA—remains underexplored. Moreover, PIRA is defined by the absence of overt clinical relapses, but this definition does not account for subclinical inflammatory activity, such as the presence of new MRI lesions. This limits the utility of PIRA as a purely non-inflammatory construct and calls for stratification into active versus non-active PIRA based on MRI findings—a distinction rarely considered in prior biomarker research.

We believe these additions more clearly convey the originality and clinical relevance of our work and appreciate the reviewer’s suggestion to improve this aspect of the manuscript.

  1. Please specify the statistical methods used for each table. Additionally, are the reported p-values adjusted or unadjusted? Given the number of categories analyzed, was any correction for multiple comparisons applied?

Response: We thank the reviewer for this helpful comment. In accordance with the journal’s formatting guidelines, we had initially provided only brief titles and footnotes. However, we have now clarified the statistical methods used in each table, and indicated whether adjustments for multiple comparisons were applied.

Specifically:

    • Table 2 presents the results from multivariable Cox regression models assessing the association between baseline variables and the risk of RAW, active PIRA, and non-active PIRA. We have updated the title to read:
      “Table 2. Multivariable Cox regression models assessing the risk of RAW, aPIRA, and naPIRA.”
    • Table 3 shows the comparison of CSF immune cell subsets and soluble factors between groups. We used analysis of variance (ANOVA) with Bonferroni correction to adjust for multiple comparisons. We have included the following footnote: “Group associations were evaluated using analysis of variance (ANOVA) with Bonferroni correction for multiple comparisons.”
    • Tables 4 and 5 present results from multivariable linear regression models, which were already specified in the respective table titles.

We confirm that the reported p-values in Table 3 are adjusted using Bonferroni correction, while those in the regression models (Tables 2, 4, and 5) are adjusted, as a result from multivariable analyses.

  1. How did the authors define and categorize the high and low levels of sNfL and sGFAP? Please provide criteria or thresholds used for grouping.

Response: For sNfL, we categorized values as high when the age- and sex-adjusted z-score was ≥1.5, in accordance with previously published studies (references 10, 12, and 14). For sGFAP, we applied age-dependent thresholds derived from our prior study on healthy controls (reference 38). Specifically, values were considered elevated when exceeding 140 pg/mL for patients under 55 years and 280 pg/mL for those aged 55 years or older.

These thresholds are detailed in the Methods section, subsection 4.8 (Definitions), and were applied consistently throughout the analyses to stratify patients into biomarker-defined groups.

  1. The introduction is somewhat insufficient and would benefit from additional background information. In particular, key concepts such as relapse-associated worsening (RAW) and progression independent of relapse activity (PIRA) should be clearly defined and discussed. Explaining why these phenomena are important and why they warrant investigation would help provide better context for the study. Additionally, the authors should consider ending the introduction with a clear statement of the study’s aim and its novelty. This will help readers understand the unique contribution of the work from the outset.

Response: We thank the reviewer for this insightful comment. We agree that further clarification of the key concepts and outcomes assessed in this study would enhance the introduction and provide better context for the reader.

The revised text (Page 3, lines 93–100) now reads (new text in bold):

“Despite these advances, the interplay between CSF and serum biomarkers in predicting the two primary pathways of disability worsening—RAW and PIRA—remains underexplored. Moreover, PIRA is defined by the absence of overt clinical relapses, but this definition does not account for subclinical inflammatory activity, such as the presence of new MRI lesions. This limits the utility of PIRA as a purely non-inflammatory construct and calls for stratification into active versus non-active PIRA based on MRI findings—a distinction rarely considered in prior biomarker research.”

We have also refined the statement of the study’s aims and its novelty to clearly position the contribution of our work (Page 3, lines 101–109) (new text in bold):

“The aim of this study was to evaluate the independent value of CSF (LS-OCMB) and serum (sNfL and sGFAP) biomarkers obtained at disease onset for predicting different types of disability worsening (RAW, active PIRA, and non-active PIRA) in a cohort of patients with MS (PwMS). By stratifying PIRA based on the presence or absence of new MRI lesions, we aimed to disentangle the inflammatory and non-inflammatory contributions to progression. Furthermore, we sought to identify distinct immunological signatures associated with these biomarkers by analyzing soluble factors and lymphocyte subsets in the CSF.”

We hope these improvements better frame the rationale, outcomes, and novelty of our study from the outset.

Reviewer 2 Report

Comments and Suggestions for Authors

In the article “Combining CSF and Serum Biomarkers to Differentiate Mechanisms of Disability Worsening in Multiple Sclerosis” by Monreal et al. the authors performed a comprehensive analysis of patients with early relapsing multiple sclerosis to investigate the prognostic value of serum and CSF biomarkers in relation to different patterns of disability progression. The study provides relevant information on the association between sNfL, sGFAP, LS-OCMB, and the risks of inflammatory and non-inflammatory worsening, offering novel insights into the biological mechanisms underlying disease progression, but several areas require further clarification.

  1. Could you please clarify the rationale for choosing a two-month exclusion period after corticosteroid administration? Are there any data confirming that this duration is sufficient to eliminate the impact on biomarkers?
  2. The frequencies of T-reg cells were associated with sNfL levels only in the presence of a positive LS-OCMB. Could you clarify whether a causal mechanism is possible here, or if this is purely a correlational observation?
  3. The multivariate linear regression included age, sex, and time from symptom onset to sampling. Were comorbidities or other inflammatory conditions that could affect sNfL and sGFAP also considered?
  4. The effect of sGFAP on the risk of naPIRA is interpreted as unrelated to inflammation. Could you clarify whether alternative sources of GFAP, such as trauma during lumbar puncture, were taken into account?
  5. There is a lack of detailed explanation of the tables in the footnotes (this may be beyond the journal’s requirements).
  6. There is a lack of detailed explanation of the figures in the footnotes (this may be beyond the journal’s requirements).
  7. In the “Materials and Methods” section, it may be possible to combine several subsections, for example in the description of the flow cytometry procedure.
  8. The authors should add catalog numbers of the antibodies used in section 4.4 of the “Materials and Methods” and include the specifications of the MRI used in section 4.7.

Conclusion:

This article presents a promising and clinically relevant concept by identifying distinct biomarker profiles associated with inflammatory and non-inflammatory disability progression in early relapsing multiple sclerosis. The integration of sNfL, sGFAP, and LS-OCMB into prognostic models offers valuable insights into disease heterogeneity and underlying mechanisms. However, to enhance the clarity and translational impact of the findings, the authors should refine the presentation of data, further justify the biomarker thresholds, and strengthen the methodological explanations regarding subgroup analyses and statistical adjustments.

Author Response

Reviewer #2

In the article “Combining CSF and Serum Biomarkers to Differentiate Mechanisms of Disability Worsening in Multiple Sclerosis” by Monreal et al. the authors performed a comprehensive analysis of patients with early relapsing multiple sclerosis to investigate the prognostic value of serum and CSF biomarkers in relation to different patterns of disability progression. The study provides relevant information on the association between sNfL, sGFAP, LS-OCMB, and the risks of inflammatory and non-inflammatory worsening, offering novel insights into the biological mechanisms underlying disease progression, but several areas require further clarification.

  1. Could you please clarify the rationale for choosing a two-month exclusion period after corticosteroid administration? Are there any data confirming that this duration is sufficient to eliminate the impact on biomarkers?

Response: The rationale for selecting a two-month washout period after corticosteroid administration was based on our prior clinical experience with IgG oligoclonal band (OCB) testing, in which we observed a lower rate of positive results when samples were obtained within two months of corticosteroid exposure, compared to samples obtained either prior to steroid use or after this interval.

Although previous studies have shown minimal or no significant impact of corticosteroids on IgG intrathecal synthesis (PMID: 11967640, 35277112, 24125567), corticosteroids are known to modulate CSF cellular populations and suppress acute inflammatory responses, which could in turn influence concentrations of dynamic biomarkers such as sNfL and sGFAP. These biomarkers are particularly sensitive to inflammatory activity and may be transiently suppressed by recent corticosteroid exposure.

To minimize this confounding effect, we conservatively excluded samples collected within two months of corticosteroid treatment. In our cohort, 40.2% of patients (n = 215) underwent sampling after receiving corticosteroids for a first demyelinating event, but only 35 patients (6.54%) had samples collected between two and three months post-treatment. This suggests that the vast majority of our biomarker measurements were obtained either before steroid use or after a sufficient interval to mitigate its potential effects.

We hope this clarifies our methodological decision and the biological rationale behind the chosen timeframe.

  1. The frequencies of T-reg cells were associated with sNfL levels only in the presence of a positive LS-OCMB. Could you clarify whether a causal mechanism is possible here, or if this is purely a correlational observation?

Response: We thank the reviewer for this insightful question. The association was significant only in the LS-OCMB-positive subgroup, which is known to exhibit a more inflammatory disease phenotype. This suggests that the observed correlation may be context-dependent and influenced by the underlying immune milieu.

In this subgroup, sNfL levels were also higher at baseline, potentially increasing the statistical power to detect correlations with immunological parameters such as T-reg frequencies. One possible interpretation is that in a highly inflammatory environment—marked by the presence of intrathecal IgM synthesis—T-regs may play a more active compensatory role in modulating neuroaxonal damage, as reflected by sNfL levels. However, this remains speculative.

Further studies with larger cohorts and mechanistic approaches will be needed to determine whether a functional relationship exists between T-reg cells, sNfL levels, and IgM OCB status.

In order to highlight this point, we have included the following modifications in the Limitations section (page 17, lines 380–392) (new text in bold):

The present study has several limitations. First, flow cytometry analyses were only conducted in a subset of the total cohort. This could limit the statistical power to detect significant differences between cohorts and requires further validation in subsequent studies. For instance, the correlation between T-reg frequencies and sNfL levels was significant only in LS-OCMB–positive patients, likely due to the higher inflammatory profile with elevated sNfL levels in this subgroup. Further studies are needed to clarify the relationship between T-reg cells, sNfL levels, and ITMS. Second, the classification of patients into aPIRA and naPIRA relied exclusively on brain MRI, as spinal cord assessments in temporal proximity to PIRA events were exceedingly rare. This reliance on MRIs may have led to an underestimation of aPIRA cases, potentially skewing results toward naPIRA. Nevertheless, the strikingly similar findings between RAW and aPIRA suggest that any underestimation of aPIRA was likely minimal and did not affect the overall conclusions.”

  1. The multivariate linear regression included age, sex, and time from symptom onset to sampling. Were comorbidities or other inflammatory conditions that could affect sNfL and sGFAP also considered?

Response: To minimize potential bias, we excluded any patient with a diagnosis other than relapsing-remitting MS (e.g., neuromyelitis optica spectrum disorder, MOG-associated disease, Sjögren’s syndrome, among others). Additionally, we ensured that no patients were sampled in the context of other conditions that could elevate biomarker levels, such as concurrent central or peripheral neurological diseases. We also excluded individuals with renal impairment, as this could artificially increase biomarker concentrations. These exclusion criteria were applied to ensure that sNfL and sGFAP levels reflected MS-related pathology as accurately as possible.

  1. The effect of sGFAP on the risk of naPIRA is interpreted as unrelated to inflammation. Could you clarify whether alternative sources of GFAP, such as trauma during lumbar puncture, were taken into account?

Response: We thank the reviewer for this important observation. All lumbar punctures were performed by trained neurologists, and any cases involving a traumatic procedure were excluded to prevent potential bias in CSF measurements. Additionally, serum samples for sNfL and sGFAP quantification were obtained at the same time as the lumbar puncture. Given the time-dependent kinetics required for biomarker elevation in serum following a traumatic event, we consider it unlikely that a traumatic lumbar puncture—if it had occurred—would have significantly influenced serum sGFAP or sNfL levels.

  1. There is a lack of detailed explanation of the tables in the footnotes (this may be beyond the journal’s requirements).

Response: We thank the reviewer for this valuable observation. We have now included the following clarifications in the footnotes to enhance the understanding of the tables:

    • Table 2:Multivariable Cox regression models including demographical (age at disease onset and sex), clinical (baseline EDSS), radiological (T2 lesion load) and biomarkers (sNfL and sGFAP levels, and LS-OCMB), including the proportion of time patients were treated with either mAbs or injectable/oral DMTs. The Harrell's C index was used as a measure of goodness of fit.
    • Table 3:Group associations were evaluated using an analysis of variance (ANOVA) with Bonferroni correction for multiple comparisons. We analyzed differences across the cohorts defined by the combination of sNfL and LS-OCMB status (columns 2 to 5), as these biomarkers were associated with inflammation-related outcomes (RAW and aPIRA). Additionally, we compared patients with high versus low sGFAP levels (columns 6 and 7), given the association of sGFAP with naPIRA.”
    • Tables 5 and 6: We believe the statistical methods and results are already sufficiently detailed in the table titles and text. However, we are happy to provide additional clarification should the reviewer or editors consider it necessary.

  1. There is a lack of detailed explanation of the figures in the footnotes (this may be beyond the journal’s requirements).

Response: In order to avoid overly extensive figure footnotes, we have aimed to keep the information concise while ensuring that the statistical methods used and the interpretation of the results are clearly reflected.

However, we have included additional clarifications:

    • Figure 1: “Figure 1. Multivariable Cox Regressions of the Risk of IAW and Non-Active PIRA. Estimation of the risk of IAW (A) and non-active PIRA (B) in patients categorized by CSF and serum biomarkers status. Results are presented as adjusted hazard ratios (HRs) with 95% confidence intervals (CIs).

aInjectable/oral DMTs: glatiramer acetate, all interferon-β formulations, fumarates, teriflunomide, sphingosine-1-phosphate receptor modulators, cladribine, or azathioprine.

bMonoclonal antibodies: Natalizumab, Alemtuzumab, Ocrelizumab, Rituximab, Ofatumumab. 

*P < 0.05; ** P < 0.001; ***P < 0.001.”.

    • Figure 2:Figure 2. Differences in CSF Regulatory T Cells Based on LS-OCMB and sNfL Status. Box plots illustrating differences in regulatory T cell (T-reg) percentages across four groups categorized by the presence or absence of LS-OCMB and high or low sNfL concentrations (A). The ROC curve showed that T-reg cells effectively distinguished between high and low sNfL levels in patients with positive LS-OCMB (B). Predictive margins derived from multivariable linear regressions (adjusted by age, sex, and time from symptom onset to sampling) demonstrate that higher T-reg percentages were associated with decreasing sNfL z-score values in patients with positive LS-OCMB (C).

*P < 0.05; **P < 0.01; ***P < 0.001.”

    • Figure 3: “Figure 3. Differences in CSF Complement C3 Levels Based on LS-OCMB and sNfL Status. Box plots showing levels in complement C3 across four groups categorized by the presence or absence of LS-OCMB and high or low sNfL levels (A). C3 levels showed potential to discriminate high sNfL z-score values in patients with negative LS-OCMB (B). Predictive margins for C3 levels were estimated from multivariable linear regressions and indicated a correlation with sNfL z-scores in patients with negative LS-OCMB (C).

*P < 0.05; **P < 0.01; ***P < 0.001. ****P < 0.0001.”

  1. In the “Materials and Methods” section, it may be possible to combine several subsections, for example in the description of the flow cytometry procedure.

Response: We thank the reviewer for this suggestion. We have combined subsections 4.4 (monoclonal antibodies) and 4.5 (surface molecule labeling) with the previous subsection (4.3. CSF analyses).

  1. The authors should add catalog numbers of the antibodies used in section 4.4 of the “Materials and Methods” and include the specifications of the MRI used in section 4.7.

Response: We have included the specifications regarding the MRI scans and antibodies used:

Page 19, line 473:MRI was performed at 1.5-T magnet (Ingenia or Achieva; Philips Healthcare) with a slide thickness of 2 to 5 mm…

Page 23, line 596: We have included a supplementary table with the specifications of the monoclonal antibodies used:

Table A2. Specifications of the monoclonal antibodies used for the analysis of CSF cells.

Monoclonal Antibody

Fluorochrome conjugated

Clone

Manufacturer

Catalog number

CD3

PerCP

SK7

BD Biosciences

345766

CD5

APC

L17F12

BD Biosciences

345783

CD8

APC-H7

SK1

BD Biosciences

641400

CD14

FITC

MφP9

BD Biosciences

345784

CD19

PE-Cy7

SJ25C1

BD Biosciences

341113

CD24

PE

ML5

BD Biosciences

555428

CD25

PE

2A3

BD Biosciences

341011

CD27

FITC

L128

BD Biosciences

340424

CD38

PE-Cy5

HIT2

BD Biosciences

555461

CD45

V500

2D1

BD Biosciences

655873

CD45RO

APC

UCHL1

BD Biosciences

559865

CD127

BV421

HIL-7R-M21

BD Biosciences

562437

CD197

PE

150503

BD Biosciences

560765

Round 2

Reviewer 1 Report

Comments and Suggestions for Authors

This manuscript is well established. Thanks for authors detailed response.